# Secretary Ranking with Minimal Inversions

**Sepehr Assadi**
Rutgers University
sepehr.assadi@rutgers.edu

**Eric Balkanski**
Harvard University
ebalkans@gmail.com

**Renato Paes Leme**
Google Research
renatoppl@google.com

## Abstract

We study a secretary problem which captures the task of ranking in online settings. We term this problem the *secretary ranking* problem: elements from an ordered set arrive in random order and instead of picking the maximum element, the algorithm is asked to assign a rank, or position, to each of the elements. The rank assigned is irrevocable and is given knowing only the pairwise comparisons with elements previously arrived. The goal is to minimize the distance of the rank produced to the true rank of the elements measured by the Kendall-Tau distance, which corresponds to the number of pairs that are inverted with respect to the true order.

Our main result is a matching upper and lower bound for the secretary ranking problem. We present an algorithm that ranks $n$ elements with only $O(n^{3/2})$ inversions in expectation, and show that any algorithm necessarily suffers $\Omega(n^{3/2})$ inversions when there are $n$ available positions. In terms of techniques, the analysis of our algorithm draws connections to linear probing in the hashing literature, while our lower bound result relies on a general anti-concentration bound for a generic balls and bins sampling process. We also consider the case where the number of positions $m$ can be larger than the number of secretaries $n$ and provide an improved bound by showing a connection of this problem with random binary trees.

## 1 Introduction

The secretary problem is one of the first problems studied in online algorithms—in fact, it was extensively studied much before the field of online algorithms even existed. It first appeared in print in 1960 as a recreational problem in Martin Gardner's Mathematical Games column in Scientific American. In the subsequent decade it caught the attention of many of the eminent probabilist researchers like Lindley [Lin61], Dynkin [Dyn63], Chow et al. [CMRS64] and Gilbert and Mosteller [GM06] among others. In a very entertaining historical survey, Ferguson [Fer89] traces the origin of the secretary problem to much earlier: Cayley in 1875 and Kepler in 1613 pose questions in the same spirit as the secretary problem.

Secretary problem has been extended in numerous directions, see for example the surveys by Sakaguchi [Sak95] and Freeman [Fre83]. The problem has had an enormous influence in computer science and has provided some of the basic techniques in the field of online and approximation algorithms. Babaioff et al extended this problem to matroid set systems [BIK07] and Knapsack [BIKK07] and perhaps more importantly, show that the secretary problem is a natural tool for designing online auctions. In the last decade, the secretary problem has also been extended to posets [KLVV11], submodular systems [BHZ10], general set systems [Rub16], stable matchings [BEF+17], non-uniform arrivals [KKN15] and applied to optimal data sampling [GD09], design of

prophet inequalities [AKW14, EHLM17], crowdsourcing systems [SM13], pricing in online settings [CEFJ14], online linear programming [AWY14] and online ad allocation [FHK$^{+}$10].

The (admittedly incomplete) list of extensions and applications in the last paragraph serves to showcase that the secretary problem has traditionally been a vehicle for deriving connections between different subfields of computer science and a testbed of new techniques.

**Ranking Secretaries.** We consider a natural variant of the secretary problem that captures ranking from pairwise comparisons in online settings. In the *secretary ranking* problem, instead of selecting the maximum element we are asked to *rank* each arriving element. In the process of deriving an optimal algorithm for this problem, we uncover novel connections between ranking and the technique of linear probing, which is one of the earliest techniques in the hashing literature studied by Knuth [Knu63], and also the expected height of random binary trees.

In the traditional secretary problem a decision maker is trying to hire a secretary. There is a total order over $n$ secretaries and the goal of the algorithm is to hire the best secretary. The secretaries are assumed to arrive in a random order and the algorithm can only observe the relative rank of each secretary with respect to the previously interviewed ones. Once a secretary is interviewed, the algorithm needs to decide whether to hire the current one or to irrevocably abandon the current candidate and continue interviewing.

In our setting, there are $m$ job positions and $n$ secretaries. There is a known total order on positions. Secretaries arrive in random order and, as before, we can only compare a secretary with previously interviewed ones. In our version, all secretaries will be hired and the decision of the algorithm is in which position to hire each secretary. Each position can be occupied by at most one secretary and hiring decisions are irrevocable. Ideally, the algorithm will hire the best secretary in the best position, the second best secretary in the second best position and so on. The loss incurred by the algorithm corresponds to the pairs that are incorrectly ordered, i.e., pairs where a better secretary is hired in a worse position.

We give two examples that illustrate scenarios where irrevocable ranking decisions occur online. The first is in the context of task assignments. For concreteness, consider a consulting firm with teams of different skill levels. Projects of different difficulty arrive in an online fashion and when a project arrives, the firm needs to decide which team will execute. Of course, the most difficult projects should go to the most skillful team. The second example is in the context of reward allocation. Consider a university department that would like to assign the best scholarships available to the best students. However, scholarships arrive one at a time and the school needs to decide which student is assigned that scholarship knowing only the relative quality of the scholarships arrived so far.

## 1.1 Our Results and Techniques

The perhaps most natural case of the secretary ranking problem is when the numbers of positions and secretaries are the same, i.e. $m = n$, which we call the dense case. The trivial algorithm that assigns a random empty position for each arriving secretary incurs $\Theta(n^2)$ cost, since each pair of elements has probability $1/2$ of being an inversion. On the other hand, $\Omega(n)$ is a trivial lower bound on the cost of any algorithm because nothing is known when the first element arrives. As such, there is a linear gap between the costs of the trivial upper and lower bounds for this secretary ranking problem. Our main result is an asymptotically tight upper and lower bound on the loss incurred by the algorithms for the secretary ranking problem.

**Theorem.** *There is an algorithm for the secretary ranking problem that computes a ranking with $\mathcal{O}(n^{3/2})$ inversions in expectation. Moreover, any algorithm for this problem makes $\Omega(n^{3/2})$ inversions in expectation.*

There are two challenges in designing an algorithm for secretary ranking. In earlier time steps, there are only a small number of comparisons observed and these do not contain sufficient information to estimate the true rank of the arriving elements. In later time steps, we observe a large number of comparisons and using the randomness of elements arrival, the true rank of the elements can be estimated well. However, the main difficulty is that at these time steps many of the positions have already been assigned to some element arrived earlier and are hence not available. The first information-theoretic challenge exacerbates this second issue. Previous bad placements might imply

that all the desired positions are unavailable for the current element, causing a large cost even for an element whose true rank is estimated accurately.

The algorithm needs to handle these two opposing challenges simultaneously. The main idea behind our algorithm is to estimate the rank of the current element using the observed comparisons and then add some noise to these estimations to obtain additional randomness in the positions and avoid positively correlated mistakes. We then assign the current element to the closest empty position to this noisy estimated rank. The main technical interest is in the analysis of this algorithm. We draw a connection to the analysis of linear probing in the hashing literature [Knu63] to argue that under this extra noise, there often exists an empty position that is close to the estimated rank.

For the lower bound, we analyze the number of random pairwise comparisons needed to estimate the rank of an element accurately. Such results are typically proven in the literature by using anti-concentration inequalities. A main technical difficulty is that most of the existing anti-concentration inequalities are for independent random variables while there is a correlation between the variables we are considering. We prove, to the best of our knowledge, a new anti-concentration inequality for a generic balls in bins problem that involves correlated sampling.

## 1.2 Related Work

Our work is inserted in the vast line of literature on the secretary problem, which we briefly discussed earlier. There has been a considerable amount of work on multiple-choice secretary problems where, instead of the single best element, multiple elements can be chosen as they arrive online [Kle05, BIKK07, BIK07, BHZ10, Rub16, KP09]. We note that in multiple-choice secretary problems, the decision at arrival of an element is still binary, whereas in secretary ranking one of $n$ positions must be chosen. More closely related to our work is a paper of Babichenko et al. [BEF$^+$17] where elements that arrive must also be assigned to a position. However, the objective is different and the goal, which uses a game-theoretic notion of stable matching, is to maximize the number of elements that are not in a blocking pair. Gobel et al. [GKT15] also studied an online appointment scheduling problem in which the goal is to assign starting dates to a set of jobs arriving online. The objective here is again different from the secretary ranking problem and is to minimize the total weight time of the jobs.

Another related line of work in machine learning is the well-known problem of learning to rank that has been extensively studied in recent years (e.g. [BSR$^+$05, CQL$^+$07, BRL07, XLW$^+$08]). Two important applications of this problem are search engines for document retrieval [L$^+$09, RJ05, LXQ$^+$07, CXL$^+$06, XL07] and collaborative filtering approaches to recommender systems [SLH10, SKB$^+$12, LY08, WRdVR08]. There has been significant interest recently in ranking from pairwise comparisons [FRPU94, BFSC$^+$13, CS15, SW17, JKSO16, HSRW16, DKMR14, BMW16, AAAK17]. To the best of our knowledge, there has not been previous work on ranking from pairwise comparisons in an online setting.

Finally, we also briefly discuss hashing, since our main technique is related to linear probing. Linear probing is a classic implementation of hash tables and was first analyzed theoretically by Knuth in 1963 [Knu63], in a report which is now regarded as the birth of algorithm analysis. Since then, different variants of this problem mainly for hash functions with limited independence have been considered in the literature [SS90, PPR07, PT10]. Reviewing the vast literature on this subject is beyond the scope of our paper and we refer the interested reader to these papers for more details.

**Organization.** The remainder of the paper is organized as follows. In Section 2 we formalize the secretary ranking problem. In Section 3, we present and analyze our algorithm. Section 4 is devoted to showing the lower bound. We discuss the case where the number of positions $m$ is different from the number of elements $n$, as well as two natural extensions of the secretary ranking problem, in Section 5. Missing proofs and standard concentration bounds are postponed to the full version of the paper as well.

## 2 Problem Setup

In the secretary ranking problem, there are $n$ elements $a_1, \ldots, a_n$ that arrive one at a time in an online manner and in a uniformly random order. There is a total ordering among the elements, but

the algorithm has only access to pairwise comparisons among the elements that have already arrived. In other words, at time $t$, the algorithm only observes whether $a_i < a_j$ for all $i, j \leq t$.

The secretary ranking problem generalizes the secretary problem in the following sense: in the secretary problem, we are only interested in finding the element with the highest rank. However, in the secretary ranking problem, the goal is to assign a rank to every arrived element and construct a complete ranking of all elements. Similar to the secretary problem, we make the enabling assumption that the order of elements arrival is uniformly random.[1]

We define the rank function $\mathsf{rk} : \{a_1, \ldots, a_n\} \rightarrow [n]$ as the true rank of the elements in the total order, i.e., $a_i < a_j$ iff $\mathsf{rk}(a_i) < \mathsf{rk}(a_j)$. Since the elements arrive uniformly at random, $\mathsf{rk}(\cdot)$ is a random permutation. Upon arrival of an element $a_t$ at time step $t$, the algorithm must, irrevocably, place $a_t$ in a position $\pi(a_t) \in [n]$ that is not yet occupied, in the sense that for $a_t \neq a_s$ we must have $\pi(a_s) \neq \pi(a_t)$. Since the main goal of the algorithm is to place the elements as to reflect the true rank as close as possible[2], we refer to $\pi(a_t)$ as the *learned rank* of $a_t$. The goal is to minimize the number of pairwise mistakes induced by the learned ranking compared to the true ranking. A pairwise mistake, or an inversion, is defined as a pair of elements $a_i, a_j$ such that $\mathsf{rk}(a_i) < \mathsf{rk}(a_j)$ according to the true underlying ranking but $\pi(a_i) > \pi(a_j)$ according to the learned ranking. We measure the cost of the algorithm in expectation over the randomness of both the arrival order of elements and the algorithm.

**Measures of sortedness.**   We point out that the primary goal in the secretary ranking problem is to learn an ordering $\pi$ of the input elements which is as close as possible to their sorted order. As such, the *cost* suffered by an algorithm is given by a *measure of sortedness* of $\pi$ compared to the true ranking. There are various measures of sortedness studied in the literature depending on the application. Our choice of using the number of inversions, also known as *Kendall's tau* measure, as the cost of an algorithm is motivated by the importance of this measure and its close connection to other measures such as *Spearman's footrule* (see, e.g., Chapter 6B in [Dia88]).

For a mapping $\pi : [n] \rightarrow [n]$, Kendall's tau $K(\pi)$ measures the number of inversions in $\pi$, i.e.:

$$K(\pi) := |\{(i, j); (\pi(a_i) - \pi(a_j))(\mathsf{rk}(a_i) - \mathsf{rk}(a_j)) < 0\}|.$$

Another important measure of sortedness is Spearman's footrule $F(\pi)$ given by: $F(\pi) := \sum_{i=1}^{n} |\mathsf{rk}(a_i) - \pi(a_i)|$, which corresponds to the summation of distances between the true rank of each element and its current position. A celebrated result of Diaconis and Graham [DG77] shows that these two measures are within a factor of two of each other, namely, $K(\pi) \leq F(\pi) \leq 2 \cdot K(\pi)$. We refer to this inequality as the DG inequality throughout the paper. Thus, up to a factor of two, the goals of minimizing the Kendall tau or Spearman's footrule distances are equivalent and, while the Kendall tau distance is used in the formulation of the problem, we also use the Spearman's footrule distance in the analysis.

## 3   The Algorithm

In this section, we describe and analyze an algorithm for the secretary ranking problem. Our main algorithmic result is the following theorem.

**Theorem 1.** *There exists an algorithm for the secretary ranking problem that incurs a cost of $O(n\sqrt{n})$ in expectation.*

In Section 4, we show that this cost incurred by the algorithm is asymptotically optimal.

### 3.1   Description of the Algorithm

The general approach behind the algorithm in Theorem 1 is as follows.

> Upon the arrival of element $a_t$ at time step $t$:
>
> 1. **Estimation step:** Estimate the true rank of the arrived element $a_t$ using the *partial* comparisons seen so far.
>
> 2. **Assignment step:** Find the nearest currently unassigned rank to this estimate and let $\pi(a_t)$ be this position.

We now describe the algorithm in more details. A natural way to estimate the rank of the $t$-th element in the estimation step is to compute the rank of this element with respect to the previous $t-1$ elements seen so far and then scale this number to obtain an estimate of the rank of this element between $1$ and $n$. However, for our analysis of the assignment step, we need to tweak this approach slightly: instead of simply rescaling and rounding, we add perturbation to the estimated rank and then round its value. This gives a nice distribution of estimated ranks which is crucial for the analysis of the assignment step. The assignment step then simply assigns a learned rank to the element as close as possible to its estimated rank. We formalize the algorithm in Algorithm 1.

---

**ALGORITHM 1:** Dense Ranking

---

1   **Input:** a set $R$ of $n$ positions, denoted here by $[n]$, and at most $n$ online arrivals.
2   **for** any time step $t \in [n]$ and element $a_t$ **do**
3      Define $r_t := |\{a_{t'} \mid a_{t'} < a_t \text{ and } t' < t\}|$.
4      Sample $x_t$ uniformly in the real interval $[r_t \cdot \frac{n}{t}, (r_t + 1) \cdot \frac{n}{t}]$ and choose $\widetilde{\mathsf{rk}}(a_t) = \lceil x_t \rceil$.
5      Set the learned rank of $a_t$ as $\pi(a_t) = \arg\min_{i \in R} \left| i - \widetilde{\mathsf{rk}}(a_t) \right|$ and remove $i$ from $R$.
6   **end**

---

We briefly comment on the runtime of the algorithm. By using any self-balancing binary search tree—such as a red-black tree or an AVL tree—to store the ranking of the arrived elements as well as the set $R$ of available ranks separately, Algorithm 1 is implementable in $O(\log n)$ time for each step, so total $O(n \log n)$ worst-case time.

We also note some similarity between this algorithm and linear probing in hashing. Linear probing is an approach to resolving collisions in hashing where, when a key is hashed to a non-empty cell, the closest neighboring cells are visited until an empty location is found for the key. The similarity is apparent to our assignment step which finds the nearest currently unassigned rank to the estimated rank of an element. The analysis of the assignment step follows similar ideas as the analysis for the linear probing hashing scheme.

### 3.2   The Analysis

The total number of inversions can be approximated within a factor of 2 by the Spearman's footrule. Therefore, we can write the cost of Algorithm 1 (up to a factor 2) as follows:

$$\sum_{t=1}^{n} |\mathsf{rk}(a_t) - \pi(a_t)| \leq \sum_{t=1}^{n} \left| \mathsf{rk}(a_t) - \widetilde{\mathsf{rk}}(a_t) \right| + \sum_{t=1}^{n} \left| \widetilde{\mathsf{rk}}(a_t) - \pi(a_t) \right|.$$

This basically breaks the cost of the algorithm in two parts: one is the cost incurred by the estimation step and the other one is the cost of the assignment step. Our analysis then consists of two main parts where each part bounds one of the terms in the RHS above. In particular, we first prove that given the partial comparisons seen so far, we can obtain a relatively good estimation to the rank of the arrived element, and then in the second part, we show that we can typically find an unassigned position in the close proximity of this estimated rank to assign to it. The following two lemmas capture each part separately. In both lemmas, the randomness in the expectation is taken over the random arrivals and the internal randomness of the algorithm:

**Lemma 3.1** (Estimation Cost). *In Algorithm 1,* $\mathbb{E}\left[ \sum_{t=1}^{n} \left| \mathsf{rk}(a_t) - \widetilde{\mathsf{rk}}(a_t) \right| \right] = O(n\sqrt{n})$.

**Lemma 3.2** (Assignment Cost). *In Algorithm 1,* $\mathbb{E}\left[ \sum_{t=1}^{n} \left| \widetilde{\mathsf{rk}}(a_t) - \pi(a_t) \right| \right] = O(n\sqrt{n})$.

Theorem 1 then follows immediately from these two lemmas and Eq (3.2). The main part of the argument is the analysis of the assignment cost, i.e., Lemma 3.2, and in particular its connection to linear probing. The analysis for the estimation cost, i.e., Lemma 3.1, follows from standard Chernoff bound arguments and is deferred to the full version of the paper.

**Assignment cost: proof of Lemma 3.2.** It is useful to think of sampling a random permutation in the following recursive way: given a random permutation over $t-1$ elements, it is possible to obtain a random permutation over $t$ elements by inserting the $t$-th element in a uniformly random position between these $t-1$ elements. Formally, given $\sigma : [t-1] \to [t-1]$, if we sample a position $i$ uniformly from $[t]$ and generate permutation $\sigma' : [t] \to [t]$ such that:

$$\sigma'(t') = \begin{cases} i & \text{if } t' = t \\ \sigma(t') & \text{if } t' < t \text{ and } \sigma'(t') < i \\ \sigma(t') + 1 & \text{if } t' < t \text{ and } \sigma'(t') > i \end{cases}$$

then $\sigma'$ will be a random permutation over $t$ elements. It is simple to see that just by fixing any permutation and computing the probability of it being generated by this process.

Thinking about sampling the permutation in this way is very convenient for this analysis since at the $t$-th step of the process, the relative order of the first $t$ elements is fixed (even though the true ranks can only be determined in the end). In that spirit, let us also define for a permutation $\sigma : [t] \to [t]$ the event $\mathcal{O}_\sigma$ that $\sigma$ is the relative ordering of the first $t$ elements:

$$\mathcal{O}_\sigma = \{a_{\sigma(1)} < a_{\sigma(2)} < \ldots < a_{\sigma(t)}\}.$$

The following proposition asserts that the randomness of the arrival and the inner randomness of the algorithm, ensures that the estimated ranks at each time step are chosen *uniformly at random* from all possible ranks in $[n]$.

**Proposition 3.3.** *The values of* $\widetilde{\mathsf{rk}}(a_1), \ldots, \widetilde{\mathsf{rk}}(a_n)$ *are i.i.d and uniformly chosen from* $[n]$.

*Proof.* First let us show that for any fixed permutation $\sigma$ over $t-1$ elements, the relative rank $r_t$ defined in the algorithm is conditionally uniformly distributed in $\{0, \ldots, t-1\}$. In other words:

$$\Pr[r_t = i \mid \mathcal{O}_\sigma] = \frac{1}{t}, \qquad \forall i \in \{0, \ldots, t-1\}.$$

Simply observe that there are exactly $t$ permutations over $t$ elements such that the permutation induced in the first $t-1$ elements is $\sigma$. Since we are sampling a random permutation in this process, each of these permutation are equally likely to happen. Moreover, since each permutation corresponds to inserting the $t$-the element in one of the $t$ positions, we obtain the above equality.

Furthermore, since the probability of each value of $r_t$ does not depend on the induced permutation $\sigma$ over the first $t-1$ elements, then $r_t$ is independent of $\sigma$. Since all the previous values $r_{t'}$ are completely determined by $\sigma$, $r_t$ is independent of all previous $r_{t'}$ for $t' < t$.

Finally observe that if $r_t$ is random from $\{0, ..., t-1\}$, then $x_t$ is sampled at random from $[0, n]$, so $\widetilde{\mathsf{rk}}(a_t)$ is sampled at random from $[n]$. Since for different values of $t \in [n]$, all $r_t$ are independent, all the values of $\widetilde{\mathsf{rk}}(a_t)$ are also independent. $\qquad \square$

Now that we established that $\widetilde{\mathsf{rk}}(a_t)$ are independent and uniform, our next task is to bound how far from the estimated rank we have to go in the assignment step, before we are able to assign a learned rank to this element. This part of our analysis will be similar to the analysis of the linear probing hashing scheme. If we are forced to let the learned rank of $a_t$ be far away from $\widetilde{\mathsf{rk}}(a_t)$, say $\left| \widetilde{\mathsf{rk}}(a_t) - \pi(a_t) \right| > k$, then this necessarily means that all positions in the integer interval $[\widetilde{\mathsf{rk}}(a_t) - k : \widetilde{\mathsf{rk}}(a_t) + k]$ must have already been assigned as a learned rank of some element. In the following, we bound the probability of such an event happening for large values of $k$ compared to the current time step $t$.

We say that the integer interval $I = [i : i + s - 1]$ of size $s$ is *popular* at time $t$, iff at least $s$ elements $a_{t'}$ among the $t-1$ elements that appear before the $t$-th element have estimated rank $\widetilde{\mathsf{rk}}(a_{t'}) \in I$.

Since by Proposition 3.3 every element has probability $s/n$ of having estimated rank in $I$ and the estimated ranks are independent, we can bound the probability that $I$ is popular using a standard application of Chernoff bound (proof deferred to the full version of the paper).

**Claim 3.4.** *Let $\alpha \geq 1$, an interval of size $s \geq 2\alpha \max\left(1, \left(\frac{t}{n-t}\right)^2\right)$ is popular at time $t$ w.p. $e^{-\Omega(\alpha)}$.*

We now use the above claim to bound the deviation between $\widetilde{\mathsf{rk}}(a_t)$ and $\pi(a_t)$. The following lemma is the key part of the argument.

**Lemma 3.5.** *For any $t \leq n$, we have $\mathbb{E}\left|\widetilde{\mathsf{rk}}(a_t) - \pi(a_t)\right| = O(\max\left(1, \left(\frac{t}{n-t}\right)^2\right))$.*

*Proof.* Fix any $\alpha \geq 1$. We claim that, if the learned rank of $a_t$ is a position which has distance at least $k_\alpha = 4\alpha \cdot \max\left(1, \left(\frac{t}{n-t}\right)^2\right)$ from its estimated rank, then necessarily there exists an interval $I$ of length at least $2k_\alpha$ which contains $\widetilde{\mathsf{rk}}(a_t)$ and is popular.

Let us prove the above claim then. Let $I$ be the shortest integer interval $[a:b]$ which contains $\widetilde{\mathsf{rk}}(a_t)$ and moreover both positions $a$ and $b$ are not assigned to a learned rank by time $t$ (by this definition, $\pi(a_t)$ would be either $a$ or $b$). For $\left|\widetilde{\mathsf{rk}}(a_t) - \pi(a_t)\right|$ to be at least $k_\alpha$, the length of interval $I$ needs to be at least $2k_\alpha$. But for $I$ to have length at least $2k_\alpha$, we should have at least $2k$ elements from $a_1, \ldots, a_{t-1}$ to have an estimated rank in $I$: this is simply because $a$ and $b$ are not yet assigned a rank by time $t$ and hence any element $a_{t'}$ which has estimated rank outside the interval $I$ is never assigned a learned rank inside $I$ (otherwise the assignment step should pick $a$ or $b$, a contradiction).

We are now ready to finalize the proof. It is straightforward that in the above argument, it suffices to only consider the integer intervals $[\widetilde{\mathsf{rk}}(a_t) - k_\alpha : \widetilde{\mathsf{rk}}(a_t) + k_\alpha]$ parametrized by the choice of $\alpha \geq 1$. By the above argument and Claim 3.4, for any $\alpha \geq 1$, we have,

$$
\begin{aligned}
\mathbb{E}\left[\left|\widetilde{\mathsf{rk}}(a_t) - \pi(a_t)\right|\right] &\leq \int_{\alpha=0}^{\infty} \Pr\left(\left|\widetilde{\mathsf{rk}}(a_t) - \pi(a_t)\right| > k_\alpha\right) \cdot k_\alpha \cdot d\alpha \\
&\leq \int_{\alpha=0}^{\infty} \Pr\left(\text{Integer interval } [\widetilde{\mathsf{rk}}(a_t) - k_\alpha : \widetilde{\mathsf{rk}}(a_t) + k_\alpha] \text{ is popular}\right) \cdot k_\alpha \cdot d\alpha \\
&\underset{\text{Claim 3.4}}{\leq} O(\max\left(1, \left(\frac{t}{n-t}\right)^2\right)) \cdot \int_{\alpha=0}^{\infty} e^{-O(\alpha)} \cdot \alpha \cdot d\alpha \\
&= O(\max\left(1, \left(\frac{t}{n-t}\right)^2\right)). \qquad \square
\end{aligned}
$$

We are now ready to finalize the proof of Lemma 3.2.

*Proof of Lemma 3.2.* We have, $\mathbb{E}\left[\sum_{t=1}^{n}\left|\widetilde{\mathsf{rk}}(a_t) - \pi(a_t)\right|\right] = \sum_{t=1}^{n} \mathbb{E}\left[\left|\widetilde{\mathsf{rk}}(a_t) - \pi(a_t)\right|\right]$ by linearity of expectation. For any $t < n/2$, the maximum term in RHS of Lemma 3.5 is 1 and hence in this case, we have $\mathbb{E}\left[\left|\widetilde{\mathsf{rk}}(a_t) - \pi(a_t)\right|\right] = O(1)$. Thus, the contribution of the first $n/2 - 1$ terms to the above summation is only $O(n)$. Also, when $t > n - \sqrt{n}$, we can simply write $\mathbb{E}\left[\left|\widetilde{\mathsf{rk}}(a_t) - \pi(a_t)\right|\right] \leq n$ which is trivially true and hence the total contribution of these $\sqrt{n}$ summands is also $O(n\sqrt{n})$. It remains to bound the total contribution of $t \in [n/2, n - \sqrt{n}]$. By Lemma 3.5, $\sum_{t=n/2}^{n-\sqrt{n}} \mathbb{E}\left[\left|\widetilde{\mathsf{rk}}(a_t) - \pi(a_t)\right|\right] \leq O(1) \cdot \sum_{t=n/2}^{n-\sqrt{n}} \left(\frac{t}{n-t}\right)^2 = O(n\sqrt{n})$, where the equality is by a simple calculation (see full version of the paper). $\qquad \square$

## 4 A Tight Lower Bound

We complement the algorithmic result from the previous section by showing that the cost incurred by the algorithm is asymptotically optimal.

**Theorem 2.** *Any algorithm for the secretary ranking problem incurs $\Omega(n\sqrt{n})$ cost in expectation.*

To prove Theorem 2, we first show that no deterministic algorithm can achieve better than $\Omega(n\sqrt{n})$ inversions and then use Yao's minimax principle to extend the lower bound to randomized algorithms (by simply fixing the randomness of the algorithm to obtain a deterministic one with the same performance over the particular distribution of the input).

The main ingredient of our proof of Theorem 2 is an anti-concentration bound for sampling without replacement which we cast as a balls in bins problem. We start by describing this balls in bin problem and prove the anti-concentration bound in Lemma 4.1. Lemma 4.2 then connects the problem of online ranking to the balls in bins problem.

To continue, we introduce some asymptotic notation that is helpful for readability. We write $v = \Theta_1(n)$ if variable $v$ is linear in $n$, but also smaller and bounded away from $n$, i.e., $v = cn$ for some constant $c$ such that $0 < c < 1$.

**Lemma 4.1.** *Assume there are $n$ balls in a bin, $r$ of which are red and the remaining $n - r$ are blue. Suppose $t < \min(r, n - r)$ balls are drawn from the bin uniformly at random without replacement, and let $\mathcal{E}_{k,t,r,n}$ be the event that $k$ out of those $t$ balls are red. Then, if $r = \Theta_1(n)$ and $t = \Theta_1(n)$, for every $k \in \{0, \ldots, t\}$: $\Pr(\mathcal{E}_{k,t,r,n}) = O(1/\sqrt{n})$.*

Our high level approach toward proving Lemma 4.1 is as follows:

1. We first use a counting argument to show that $\Pr(\mathcal{E}_{k,t,r,n}) = \binom{r}{k}\binom{n-r}{t-k}/\binom{n}{t}$.

2. We then use Stirling's approximation to show $\binom{r}{k}\binom{n-r}{t-k}/\binom{n}{t} = O(n^{-1/2})$ for $k = \lfloor \frac{tr}{n} \rfloor$.

3. Finally, with a max. likelihood argument, we show that $\arg\max_{k\in[n]} \binom{r}{k}\binom{n-r}{t-k}/\binom{n}{t} \approx \frac{tr}{n}$.

By combining these, we have, $\Pr(\mathcal{E}_{k,t,r,n}) \leq \max_{k\in[n]} \binom{r}{k}\binom{n-r}{t-k}/\binom{n}{t} \leq \binom{r}{k^*}\binom{n-r}{t-k^*}/\binom{n}{t}$ for $k^* \approx \frac{tr}{n}$ (by the third step), which we bounded by $O(n^{-1/2})$ (in the second step). The actual proof is however rather technical and is postponed to the full version of the paper.

The next lemma shows that upon arrival of $a_t$, any position has probability at least $O(1/\sqrt{n})$ of being the correct rank for $a_t$, under some mild conditions. The proof of this lemma uses the previous anti-concentration bound for sampling without replacement by considering the elements smaller than $a_t$ to be the red balls and the elements larger than $a_t$ to be the blue balls. For $a_t$ to have rank $r$ and be the $k$th element in the ranking so far, the first $t - 1$ elements previously observed must contain $k - 1$ red balls out of the $r - 1$ red balls and $t - k$ blue balls out of the $n - r$ blue balls.

**Lemma 4.2.** *Fix any permutation $\sigma$ of $[t]$ and let $\mathcal{O}_\sigma$ denote the event that $a_{\sigma(1)} < a_{\sigma(2)} < \ldots < a_{\sigma(t)}$. If $\sigma(k) = t$, $k = \Theta_1(t)$ and $t = \Theta_1(n)$ then for any $r$: $\Pr(\mathsf{rk}(a_t) = r \mid \mathcal{O}_\sigma) = O(1/\sqrt{n})$.*

*Proof.* Define $\mathcal{E}_k$ as the event that "$a_t$ is the $k$-th smallest element in $a_1, \ldots, a_t$". We first have, $\Pr(\mathsf{rk}(a_t) = r \mid \mathcal{O}_\sigma) = \Pr(\mathsf{rk}(a_t) = r \mid \mathcal{E}_k)$. This is simply because $\mathsf{rk}(a_t)$ is only a function of the pairwise comparisons of $a_t$ with other elements and does not depend on the ordering of the remaining elements between themselves. Moreover,

$$\Pr(\mathsf{rk}(a_t) = r \mid \mathcal{E}_k) = \Pr(\mathcal{E}_k \mid \mathsf{rk}(a_t) = r) \cdot \frac{\Pr(\mathsf{rk}(a_t) = r)}{\Pr(\mathcal{E}_k)} = \Pr(\mathcal{E}_k \mid \mathsf{rk}(a_t) = r) \cdot \frac{t}{n}$$

since $a_t$ is randomly partitioned across the $[n]$ elements. Notice now that conditioned on $\mathsf{rk}(a_t) = r$, the event $\mathcal{E}_k$ is exactly the event $\mathcal{E}_{k-1,t-1,r-1,n-1}$ in the sampling without replacement process defined in Lemma 4.1. The $n - 1$ balls are all the elements but $a_t$, the $r - 1$ red balls correspond to elements smaller than $a_t$, the $n - r$ blue balls to elements larger than $a_t$, and $t - 1$ balls drawn are the elements arrived before $a_t$. Finally, observe that $\Pr(r < k|\mathcal{E}_k) = 0$, so for $r < k$, the bound holds trivially. In the remaining cases, $r = \Theta_1(n)$ and we use the bound in Lemma 4.1 with $t/n = \Theta(1)$. □

Using the previous lemma, we can lower bound the cost due to the $t$-th element. Fix any deterministic algorithm $\mathcal{A}$ for the online ranking problem. Recall that $\pi(a_t)$ denotes the learned rank of the item $a_t$ arriving in the $t$-th time step. For any time step $t \in [n]$, we use $\mathsf{cost}_\mathcal{A}(t)$ to denote the cost incurred by the algorithm $\mathcal{A}$ in positioning the item $a_t$. More formally, if $\mathsf{rk}(a_t) = i$, we have $\mathsf{cost}_\mathcal{A}(t) := |i - \pi(a_{(t)})|$. Theorem 2 then follows by Yao's minimax principle principle and the following lemma, whose proof appears in the appendix

**Lemma 4.3.** *Fix any deterministic algorithm $\mathcal{A}$. For any $t = \Theta_1(n)$, $\mathbb{E}[\mathsf{cost}_\mathcal{A}(t)] = \Omega(\sqrt{n})$.*

# 5 Additional Results and Extensions

In the full version of the paper, we study two additional cases of the secretary ranking problem.

- In the sparse case, we wish to compute how large the number $m$ of positions needs to be such that we incur no inversions. Clearly for $m = 2^{n+1} - 1$ it is possible to obtain zero inversions with probability 1 and for any number less than that it is also clear that any algorithm needs to cause inversions with non-zero probability. If we only want to achieve zero inversions with high probability, how large does $m$ need to be? By showing a connection between the secretary problem and random binary trees, we show that for $m \geq n^\alpha$ for $\alpha \approx 2.998$ it is possible to design an algorithm that achieves zero inversion with probability $1 - o(1)$. The constant $\alpha$ here is obtained using the high probability bound on the height of a random binary tree of $n$ elements.

- We combine the algorithms for the dense and sparse cases to obtain a general algorithm with a bound on the expected number of inversions which smoothly interpolates between the bounds obtained for the dense and sparse cases. This algorithm starts by running the algorithm for the sparse case and when two elements are placed very closed to each other by the sparse algorithm, we switch to use the algorithm for the dense case to assign a position to remaining elements with rank between these two close elements.

We discuss two extensions of the secretary ranking problem that we will include in the full version of the paper.

- **Number of comparisons.** A first extension is to minimize the number of pairwise comparisons between elements needed to obtain a small number of inversions. It is possible to obtain $O(n^{3/2})$ inversions with only $O(n \log n)$ comparisons, which is optimal up to lower order terms. Our algorithm maintains an ordering of the elements seen so far. Thus, when a new element arrives, the algorithm can find the position of this new element in the current ordering by binary search.

- **Noisy comparisons.** Our main result extends to a noisy model where each comparison is, independently, correct with probability $1 - \epsilon$ and wrong with probability $\epsilon$. The impact of the noise in the analysis is an additional $\epsilon n$ additive term per element in the estimation cost and no additional term for the assignment cost. The total cost is then $O(n^{3/2} + \epsilon n^2)$.

  The reason this can be done is that the analysis is modular and decomposes the error in an estimation and assignment terms. If we estimate the new rank using line 3 of Algorithm 1 but with noisy comparisons, we still have an unbiased estimator of the rank (but with larger variance) and Proposition 3.3 still holds. Those are the only requirements for the analysis of the assignment cost, so it remains unchanged.

If we have noisy comparisons and also want to improve the number of comparisons, the technique in (a) for reducing the number of comparisons no longer holds, but we can still reduce the number of comparisons by estimating $\tilde{r}$ in line 3 of Algorithm 1 by sampling only a few of the previous elements and paying the corresponding cost in increased variance. The same analysis as in (b) still holds.

## Footnotes

[1]It is straightforward to verify that when the ordering is adversarial, any algorithm incurs the trivial cost of $\Omega(n^2)$. For completeness, a proof is provided in the full version of the paper.

[2]In other words, hire the better secretaries in better positions.

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
