[Supplementary Material]

# Secretary Ranking with Minimal Inversions

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

\right) \cdot \frac{\Pr\left(\mathsf{rk}(a_t) = r\right)}{\Pr\left(\mathcal{E}_k\right)} = \Pr\left(\mathcal{E}_k \mid \mathsf{rk}(a_t) = r\right) \cdot \frac{t}{n}$$

since $a_t$ is randomly partitioned across the $[n]$ elements. Notice now that conditioned on $\mathsf{rk}(a_t) = r$, the event $\mathcal{E}_k$ is exactly the event $\mathcal{E}_{k-1,t-1,r-1,n-1}$ in the sampling without replacement process defined in Lemma 4.1. The $n - 1$ balls are all the elements but $a_t$, the $r - 1$ red balls correspond to elements smaller than $a_t$, the $n - r$ blue balls to elements larger than $a_t$, and $t - 1$ balls drawn are the elements arrived before $a_t$. Finally, observe that $\Pr\left(r < k|\mathcal{E}_k\right) = 0$, so for $r < k$, the bound holds trivially. In the remaining cases, $r = \Theta_1(n)$ and we use the bound in Lemma 4.1 with $t/n = \Theta(1)$. □

Using the previous lemma, we can lower bound the cost due to the $t$-th element. Fix any deterministic algorithm $\mathcal{A}$ for the online ranking problem. Recall that $\pi(a_t)$ denotes the learned rank of the item $a_t$ arriving in the $t$-th time step. For any time step $t \in [n]$, we use $\mathsf{cost}_{\mathcal{A}}(t)$ to denote the cost incurred by the algorithm $\mathcal{A}$ in positioning the item $a_t$. More formally, if $\mathsf{rk}(a_t) = i$, we have $\mathsf{cost}_{\mathcal{A}}(t) := |i - \pi(a_{(t)})|$. Theorem 2 then follows by Yao's minimax principle principle and the following lemma, whose proof appears in the appendix

**Lemma 4.3.** *Fix any deterministic algorithm $\mathcal{A}$. For any $t = \Theta_1(n)$, $\mathbb{E}\left[\mathsf{cost}_{\mathcal{A}}(t)\right] = \Omega\left(\sqrt{n}\right)$.*

## Footnotes

[1]In other words, hire the better secretaries in better positions.

[2]It is straightforward to verify that when the ordering is adversarial, any algorithm incurs the trivial cost of

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

## A   Useful Concentration of Measure Inequalities

We use the following two standard versions of Chernoff bound (see, e.g., [DP09]) throughout.

**Proposition A.1** (Multiplicative Chernoff bound). *Let $X_1, \ldots, X_n$ be $n$ independent random variables taking values in $[0, 1]$ and let $X := \sum_{i=1}^{n} X_i$. Then, for any $\varepsilon \in (0, 1]$,*

$$\Pr\left(X \geq (1 + \varepsilon) \cdot \mathbb{E}\left[X\right]\right) \leq \exp\left(-2\varepsilon^2 \cdot \mathbb{E}\left[X\right]\right).$$

**Proposition A.2** (Additive Chernoff bound). *Let $X_1, \ldots, X_n$ be $n$ independent random variables taking values in $[0, 1]$ and let $X := \sum_{i=1}^{n} X_i$. Then,*

$$\Pr\left(|X - \mathbb{E}\left[X\right]| > t\right) \leq 2 \cdot \exp\left(-\frac{2t^2}{n}\right).$$

*Moreover, if $X_1, \ldots, X_n$ are* negatively correlated *(i.e. $\Pr[X_i = 1, \forall i \in S] \leq \prod_{i \in S} \Pr[X_i = 1]$ for all $S \subseteq [n]$), then the upper tail holds:* $\Pr\left(X - \mathbb{E}\left[X\right] > t\right) \leq \exp\left(-\frac{2t^2}{n}\right)$.

Moreover, in the above setting, if $X$ comes from a sampling *with* replacement process, then the inequality holds for both upper and lower tails. For sampling without replacement, we refer to Serfling [Ser74] for a complete discussion and for Chernoff bounds for negatively correlated random variables see [PS97].

**Proposition A.3** (Chernoff bound for sampling without replacement). *Consider an urn with $a \geq b$ red and blue balls. Draw $b$ balls uniformly from the urn without replacement and let $X$ be the number of red balls drawn, then the two sided bound holds:* $\Pr\left(|X - \mathbb{E}\left[X\right]| > t\right) \leq 2 \cdot \exp\left(-\frac{2t^2}{b}\right)$.

*Proof.* If $X_i$ is the event that the $i$-th ball is red, then since $X_i$ are negatively correlated, the upper tail Chernoff bound of $X = \sum_i X_i$ holds. Now, let $Y_i = 1 - X_i$ be the probability that the the $i$-th ball is blue and $Y = \sum_i Y_i$. The upper tail for $Y$ correspond to the lower tail for $X$, i.e.:

$\Pr\left(X - \mathbb{E}\left[X\right] < t\right) = \Pr\left(Y - \mathbb{E}\left[Y\right] > t\right) \leq \exp\left(-\frac{2t^2}{b}\right).$ $\qquad\qquad\square$

## B   Sparse Secretary Ranking

In this section, we consider the special case where the number of positions is very large, which we call sparse secretary ranking. In the extreme when $m \geq 2^{n+1} - 1$ it is possible to assign a position to each secretary without ever incurring a mistake. To do that, build a complete binary tree of height $n$ and associate each position in $[m]$ with a node (both internal and leaf) of the binary tree such that the order of the positions corresponds to the pre-order induced by the binary tree (see figure B). Once the elements arrive in an online fashion, insert them in the binary tree and allocate them in the corresponding position.

Figure 1: Illustration of the binary tree algorithm for $m = 7$ and order $a_2 < a_3 < a_1$.

We note that the above algorithm works for any order of arrival. If the elements arrive in random order, it is possible to obtain zero inversions with high probability for an exponentially smaller value of $m$. The idea is very similar to the one outlined above. Let $H_n$ be a random variable corresponding to the height of a binary tree built from $n$ elements in random order. Reed [Ree03] shows that

$\mathbb{E}[H_n] = \alpha \ln(n)$, $\text{Var}[H_n] = O(1)$ where $\alpha$ is the solution of the equation $\alpha \ln(2e/\alpha) = 1$ which is $\alpha \approx 4.31107$.

Since the arrival order of secretaries is uniformly random, the binary tree algorithm won't touch any node with height more than $\bar{h} = \lceil(\alpha + O(\epsilon)) \ln(n)\rceil$ with probability $1 - o(1)$. This observation allows us to define an algorithm that obtains zero inversions with probability $1 - o(1)$. If $m \geq 2^{\bar{h}+1} - 1 = \Omega(n^{2.998+\epsilon})$, we can build a binary tree with height $\bar{h}$ and associate each node of the tree to a position. Once the elements arrive, allocate the item in the corresponding position. If an item is added to the tree with height larger than $\bar{h}$, start allocating the items arbitrarily.

**Theorem 3.** *If $m \geq n^{2.988+\epsilon}$ then the algorithm that allocates according to a binary tree incurs zero inversions with probability $1 - o(1)$.*

Devroye [Dev86] bounds the tail of the distribution of $H_n$ as follows:

$$\Pr[H_n \geq k \cdot \ln n] \leq \frac{1}{n} \cdot \left(\frac{2e}{k}\right)^{k \cdot \ln n}$$

for $k > 2$. In particular: $\Pr[H_n \geq 6.3619 \cdot \ln n] \leq 1/n^2$. Adapting the analysis above, we can show that for $m \geq 4.41$ (where $4.41 = 6.3619 \cdot \ln(2)$) the algorithm incurs less than one inversion in expectation.

**Corollary 4.** *If $m \geq \Omega(n^{4.41})$ then the algorithm that allocates according to a binary tree incurs $O(1)$ inversion in expectation.*

# C General Secretary Ranking

In the general case, we combine the ideas for the sparse and dense case to obtain an algorithm interpolating both cases. As described in Algorithm 2, we construct a complete binary search tree of height $h$ and associating one position for each internal node, but for the leaves we associate a block of $w = m/2^h - 1$ positions (see Figure 2). If we insert an element in a leaf, we allocate according to an instance of the dense ranking algorithm. By that we mean that the algorithm pretends that the elements allocated to that leaf are an isolated instance of dense ranking with $w$ elements and $w$ positions. We will set $h$ such that in expectation there only $w$ elements in each leaf with high probability. If at some point more than $w$ elements are placed in any given leaf, the algorithm starts allocating arbitrarily.

Figure 2: Illustration of the general algorithm (Algorithm 2) for order $a_2 < a_3 < a_4 < a_1$. The leaves are associated with blocks of $w$ consecutive positions and internal nodes are associated with a single position. Elements $a_3$ and $a_4$ are associated with the same leaf and therefore we place them in a block of $w$ positions as we would in a dense ranking problem with $w$ arrivals and $w$ positions.

---

**ALGORITHM 2:** General secretary ranking

---

1  **Input:** a set of $m$ positions, at most $n$ online arrivals and a height $h$.
2  Construct a complete binary search tree $T$ of height $h$ and associate one position for each internal node, and
     a block of $w = m/2^h - 1$ positions for each leaf such that the order of the positions corresponds to the
     pre-order induced by the binary tree
3  **for** any time step $t \in [n]$ and element $a_t$ **do**
4       Insert $a_t$ in the tree $T$
5       **if** $a_t$ reaches an empty internal node **then**
6          Place $a_t$ in the position corresponding to this internal node
7       **end**
8       **else**
9          Place $a_t$ according to an instance of the dense ranking algorithm (Algorithm 1) over the block of
             positions corresponding to the leaf reached by $a_t$. If there are no position available in that block,
             place $a_t$ arbitrarily
10      **end**
11 **end**

---

For stating our main theorem and its proof, it is convenient to define the functions:

$$f(\alpha) = \frac{\alpha \ln(2) - 1}{1 - 2\alpha \ln(2e/\alpha)} \qquad g(\alpha) = \frac{1}{1 - 2\alpha \ln(2e/\alpha)}$$

defined in the interval $(\alpha_0, \infty)$ where $\alpha_0 \approx 4.910$ is the solution to the equation $1 - 2\alpha_0 \ln(2e/\alpha_0) = 0$. Both functions are monotone decreasing from $+\infty$ (when $\alpha = \alpha_0$) to zero (when $\alpha \to \infty$). We are now ready to state our main theorem:

**Theorem 5.** *Assume $m \geq 10n \log n$ and let $\alpha \in (\alpha_0, \infty)$ be the solution to $\frac{m}{9n \log n} = n^{f(\alpha)}$, then the expected number of inversions of the general secretary ranking algorithm with $h = \alpha \ln(n^{g(\alpha)})$ is $\tilde{O}(n^{1.5 - 0.5g(\alpha)})$.*

We note that the algorithm smoothly interpolated between the two cases previously analyzed. When $m = n\log(n)$ then $\alpha \to \infty$, so $g(\alpha) \to 0$ and the bound on the theorem becomes $\tilde{O}(n^{1.5})$. In the other extreme, when $m \to \infty$, then $\alpha \to \alpha_0$ and therefore $g(\alpha) \to \infty$, so the bound on the number of inversions becomes $O(1)$.

*Proof.* Let $H_t$ be the height of the binary tree formed by the first $t$ elements. By Devroye's bound [Dev86], the probability that a random binary tree formed by the first $t := n^{g(\alpha)}$ elements has height more than $h = \alpha \ln(t)$ is

$$\Pr[H_t \geq h] \leq \frac{1}{t}(2e/\alpha)^{\alpha \ln t} = t^{\alpha \ln(2e/\alpha) - 1}.$$

In case this event happens, we will use the trivial bound of $O(n^2)$ on the number of inversions, which will contribute

$$n^2 t^{\alpha \ln(2e/\alpha) - 1} = n^{1.5 - 0.5/(1 - 2\alpha \ln(2e/\alpha))} = n^{1.5 - 0.5g(\alpha)}$$

to the expectation. From this point on, we consider the remaining event that $H_t < h$.

Next, we condition on the first $t$ elements that we denote $b_1, \ldots, b_t$ such that $b_1 < \cdots < b_t$. We note that for each remaining element $a_i$, $i > t$, we have $b_j < a_i < b_{j+1}$ with probability $1/(t+1)$ for all $j \in [t]$. Since $b_1, \ldots, b_t$ are all placed in positions corresponding to internal nodes, each element has at most probability $1/t$ of hitting any of the dense-ranking instances. Thus, each dense ranking instance receives at most $n/t$ elements in expectation, and by a standard application of the Chernoff bound, the probability that a dense ranking instance sees more than $9(n/t) \log n$ elements is $n^{-3}$. If this is the case for some dense ranking instance, we again use the $n^2$ trivial bound, which contributes at most 1 to the expected number of inversions. For the remainder of the proof, we assume that each dense ranking instance gets at most $9(n/t) \log n$ elements.

Next, note that the size of each block is

$$w = \frac{m}{2^h} - 1 = \frac{m}{t^{\alpha \ln(2)}} - 1 \geq 9(n/t) \log n$$

where the last equality is by definition of $t$. Thus, no more than $w$ elements are inserted in any leaf.

Let $v_i$ is the number of elements in each of the dense rank instances. We note that within the elements in each dense ranking block the arrival order is random, so we can apply the bound from Section 3 and obtain by Theorem 1 that the total expected cost from the inversions caused by dense rank is at most

$$\sum_i O\left(v_i^{1.5}\right) \leq \tilde{O}(t \cdot (n/t)^{1.5}) = \tilde{O}(n^2 t^{\alpha \ln(2e/\alpha)-1}) = \tilde{O}(n^{1.5-0.5g(\alpha)})$$

since $\sum_i v_i = n$ and $v_i \leq (n/t)\log(n)$. By the construction there are no inversions between elements inserted in different leaves and between an element inserted in an internal node and any other element. Summing the expected number of mistakes from the events $H_t \geq h$ and $H_t < h$, we get the bound in the statement of the theorem. $\qquad\square$

# D  Missing Analysis from Section 3

**Estimation Cost: Proof of Lemma 3.1.**  We begin with the following useful proposition.

**Proposition D.1.** *If* $1 < t \leq n$ *and* $0 \leq r \leq t-1$, *then* $r \cdot \left(\frac{n}{t}\right) \leq r \cdot \left(\frac{n-1}{t-1}\right) \leq (r+1) \cdot \left(\frac{n}{t}\right)$.

*Proof.* $0 \leq r\left(\frac{n-1}{t-1} - \frac{n}{t}\right) = r\frac{n-t}{t(t-1)} \leq (t-1)\frac{n-t}{t(t-1)} \leq \frac{n}{t}$. $\qquad\square$

The correctness of the estimation step in our algorithm relies on the following proposition that bounds the probability of the deviation between the estimated rank and the true rank of each element (depending on the time step it arrives). The proof uses the Chernoff bound for sampling without replacement.

**Proposition D.2.** *For any* $t > 1$ *and any* $\alpha \geq 0$, $\Pr\left(\left|\mathsf{rk}(a_t) - \widetilde{\mathsf{rk}}(a_t)\right| \geq 1 + \frac{n}{t} + \alpha \cdot \frac{n-1}{\sqrt{t-1}}\right) \leq e^{-\Omega(\alpha^2)}$.

*Proof.* Fix any $t \in [n]$ and element $a_t$ and recall that $\mathsf{rk}(a_t)$ denotes the true rank of $a_t$. Conditioned on a fixed value for the rank of $a_t$, the distribution of the number of elements $r_t$ that arrived before $a_t$ and have a smaller rank is equivalent to a sampling without replacement process of $t-1$ balls where the urn has $\mathsf{rk}(a_t) - 1$ red balls and $n - \mathsf{rk}(a_t)$ blue balls (and the goal is to count the number of red balls). As such $\mathbb{E}[r_t] = \frac{\mathsf{rk}(a_t)-1}{n-1}$ and by the Chernoff bound for sampling without replacement (Proposition A.3 with $a = n$ and $b = t-1$), we have:

$$\Pr\left(|r_t - \mathbb{E}[r_t]| \geq \alpha\sqrt{t-1}\right) \leq 2 \cdot \exp\left(-\frac{2(\alpha\sqrt{t-1})^2}{t-1}\right) = e^{-\Omega(\alpha^2)}.$$

We now argue that

$$\Pr\left(\left|\mathsf{rk}(a_t) - \widetilde{\mathsf{rk}}(a_t)\right| \geq 1 + \frac{n}{t} + \alpha \cdot \frac{n-1}{\sqrt{t-1}}\right) \leq \Pr\left(|r_t - \mathbb{E}[r_t]| \geq \alpha\sqrt{t-1}\right).$$

which finalizes the proof by the bound in above equation.

To see this, note that,

$$\alpha\frac{n-1}{\sqrt{t-1}} \geq \left|\frac{n-1}{t-1}r_t - \mathsf{rk}(a_t)\right| \geq |x_t - \mathsf{rk}(a_t)| - \frac{n}{t} \geq \left|\widetilde{\mathsf{rk}}(a_t) - \mathsf{rk}(a_t)\right| - 1 - \frac{n}{t}$$

The first inequality follows from substituting the expectation in $|r_t - \mathbb{E}[r_t]| \geq \alpha\sqrt{t-1}$ and multiplying the whole expression by $(n-1)/(t-1)$. The second inequality just follows from the fact that both the variable $x_t$ (defined in step 4 of Algorithm 2) and $\frac{n-1}{t-1}r_t$ are in the interval $[\frac{n}{t}r_t, \frac{n}{t}(r_t+1)]$. The fact that $x_t$ is in this interval comes directly from its definition in the algorithm and the fact that $\frac{n-1}{t-1}r_t$ is in the interval is by a simple calculation (see Proposition D.1 in Appendix D). The last inequality follows from the fact that $\widetilde{\mathsf{rk}}(a_t) = \lceil x_t \rceil$. $\qquad\square$

600 We are now ready to prove Lemma 3.1.

601 *Proof of Lemma 3.1.* Fix any $t > 1$; we have,

$$\mathbb{E}\left[\left|\mathsf{rk}(a_t) - \widetilde{\mathsf{rk}}(a_t)\right| - 1 - \frac{n}{t}\right] \leq \int_{\alpha=0}^{\infty} \Pr\left(\left|\mathsf{rk}(a_t) - \widetilde{\mathsf{rk}}(a_t)\right| - 1 - \frac{n}{t} \geq \alpha \cdot \frac{n-1}{\sqrt{t-1}}\right) \cdot \frac{n-1}{\sqrt{t-1}} \cdot d\alpha$$

$$\leq \frac{n-1}{\sqrt{t-1}} \cdot \int_{\alpha=0}^{\infty} e^{-\Omega(\alpha^2)} \cdot d\alpha = O\left(\frac{n}{\sqrt{t}}\right). \quad \text{(by Proposition D.2)}$$

602 Hence, using the trivial bound for $t = 1$ and the bound above for $t > 1$ we conclude that:

$$\mathbb{E}\left[\sum_{t=1}^{n}\left|\mathsf{rk}(a_t) - \widetilde{\mathsf{rk}}(a_t)\right|\right] = \sum_{t=1}^{n} \mathbb{E}\left[\left|\mathsf{rk}(a_t) - \widetilde{\mathsf{rk}}(a_t)\right|\right] = \sum_{t=1}^{n} O\left(\frac{n}{t} + \frac{n}{\sqrt{t}}\right) = O(n\sqrt{n}) \quad \square$$

603 **Missing analysis for Lemma 3.2.**

604 **Claim 3.4.** *Let* $\alpha \geq 1$*, an interval of size* $s \geq 2\alpha \max\left(1, \left(\frac{t}{n-t}\right)^2\right)$ *is popular at time* $t$ *w.p.* $e^{-O(\alpha)}$.

605 *Proof.* The proof follows directly from the Chernoff bound in Proposition A.1. For $t' \in [t]$, let $X_{t'}$
606 be the event that $\widetilde{\mathsf{rk}}(a_{t'}) \in I$ and $X = \sum_{t'=1}^{t} X_{t'}$, then setting $\epsilon = \min(1, \frac{n-t}{t})$ we have that:

$$\Pr\left(I \text{ is popular}\right) = \Pr\left(X \geq s\right) \leq \Pr\left(X > (1+\varepsilon) \cdot \epsilon \cdot \mathbb{E}[X]\right)$$

$$\leq \exp\left(-\frac{\varepsilon^2 \cdot \mathbb{E}[X]}{2}\right) = e^{-O(\alpha)}$$

607 as $\mathbb{E}[X] = s \cdot t/n$. $\square$

608 **Proposition D.3.** *For any integer* $n > 0$, $\sum_{t=1}^{n-\sqrt{n}}\left(\frac{t}{n-t}\right)^2 = O(n\sqrt{n})$.

609 *Proof.* By defining $k = n - t$, we have,

$$\sum_{t=1}^{n-\sqrt{n}}\left(\frac{t}{n-t}\right)^2 = \sum_{k=\sqrt{n}}^{n-1}\left(\frac{n-k}{k}\right)^2 \leq \sum_{k=\sqrt{n}}^{n-1}\left(\frac{n}{k}\right)^2$$

610 For $i \in [\sqrt{n}]$, define $K_i := \{k \mid i \cdot \sqrt{n} \leq k < (i+1) \cdot \sqrt{n}\}$. For any $k \in K_i$, we have, $\frac{n}{k} \leq \frac{\sqrt{n}}{i}$.
611 As such, we can write,

$$\sum_{k=\sqrt{n}}^{n-1}\left(\frac{n}{k}\right)^2 = \sum_{i=1}^{\sqrt{n}}\sum_{k \in K_i}\left(\frac{n}{k}\right)^2 \leq \sum_{i=1}^{\sqrt{n}}\sum_{k \in K_i}\left(\frac{\sqrt{n}}{i}\right)^2$$

$$\leq \sum_{i=1}^{\sqrt{n}} n \cdot |K_i| \cdot \frac{1}{i^2} \leq n\sqrt{n} \cdot \sum_{i=1}^{\sqrt{n}} \frac{1}{i^2} = O(n\sqrt{n})$$

612 as the series $\sum_i \frac{1}{i^2}$ is a converging series. $\square$

## E Anti-Concentration for Sampling Without Replacement

614 We prove Lemma 4.1 restated here for convenience.

615 **Lemma** (Restatement of Lemma 4.1). *Assume there are* $n$ *balls in a bin,* $r$ *of which are red and the*
616 *remaining* $n - r$ *are blue. Suppose* $t < \min(r, n - r)$ *balls are drawn from the bin uniformly at*
617 *random without replacement, and let* $\mathcal{E}_{k,t,r,n}$ *be the event that* $k$ *out of those* $t$ *balls are red. Then, if*
618 $r = \Theta_1(n)$ *and* $t = \Theta_1(n)$*, for every* $k \in \{0, \dots, t\}$: $\Pr\left(\mathcal{E}_{k,t,r,n}\right) = O\left(1/\sqrt{n}\right)$.

619 To prove Lemma 4.1, we will describe the sampling without replacement process explicitly and
620 bound the relevant probabilities.

**Proposition E.1.** *Let $0 < c < 1$ be a constant. Then:*

$$\binom{n}{cn} = \Theta(n^{-1/2}c^{-(cn+1/2)}(1-c)^{-((1-c)n+1/2)})$$

The notation $y = \Theta(x)$ in the lemma statement means that there are universal constants $0 < \underline{\alpha} < \overline{\alpha}$ independent of $c$ and $n$ such that $\underline{\alpha} \cdot x \leq y \leq \overline{\alpha} \cdot x$. The proof is based on the following version of Stirling's approximation: $\sqrt{2\pi}\, n^{n+\frac{1}{2}}e^{-n} \leq n! \leq e\, n^{n+\frac{1}{2}}e^{-n}$. which can be written in our notation as: $n! = \Theta(n^{n+\frac{1}{2}}e^{-n})$. The proof of the previous lemmas follows from just expanding the factorials in the definition of the binomial:

*Proof.* Observe that

$$\binom{n}{cn} = \frac{n!}{(cn)!((1-c)n)!} = \Theta\left(\frac{n^{n+\frac{1}{2}}e^{-n}}{(cn)^{cn+\frac{1}{2}}e^{-cn}((1-c)n)^{(1-c)n+\frac{1}{2}}e^{-((1-c)n)}}\right)$$

The statement follows from simplifying the right hand side. $\square$

**Lemma E.2.** *Assume that $r = \Theta_1(n)$ and $t = \Theta_1(n)$ and $t \leq \min(r, n-r)$, then for $k = \lfloor rt/n \rfloor$, we have*

$$\frac{\binom{r}{k} \cdot \binom{n-r}{t-k}}{\binom{n}{t}} = \mathcal{O}\left(1/\sqrt{n}\right).$$

*Proof.* Start by writing $r = c_r \cdot n$ and $t = c_t \cdot n$ for $0 < c_r, c_t < 1$. It will be convenient to assume that $k = rt/n$ is an integer (if not and we need to apply floors, the exact same proof work by keeping track of the errors introduced by floor). Then we can write: First, note that

$$\frac{\binom{r}{k} \cdot \binom{n-r}{t-k}}{\binom{n}{t}} = \frac{\binom{c_r n}{c_t c_r n} \cdot \binom{(1-c_r)n}{(1-c_r)c_t n}}{\binom{n}{c_t n}}$$

We can now apply the approximation in Proposition E.1 obtaining:

$$\Theta\left(\frac{n^{1/2}c_t^{c_t n+\frac{1}{2}}(1-c_t)^{(1-c_t)n+\frac{1}{2}}}{(c_r n)^{1/2}c_t^{c_t(c_r n)+\frac{1}{2}}(1-c_t)^{(1-c_t)(c_r n)+\frac{1}{2}}((1-c_r)n)^{1/2}c_t^{c_t((1-c_r)n)+\frac{1}{2}}(1-c_t)^{(1-c_t)((1-c_r)n)+\frac{1}{2}}}\right)$$

Simplifying this expressoin, we get: $\Theta\left((nc_t(1-c_t)c_r(1-c_r))^{-1/2}\right) = \Theta_1(1/\sqrt{n})$. $\square$

**Lemma E.3.** *Fix any $r, t, n$ such that $r, t \leq n$. Then,*

$$\arg\max_{k\in[n]} \frac{\binom{r}{k} \cdot \binom{n-r}{t-k}}{\binom{n}{t}} = \left\lfloor t \cdot \frac{r}{n} \right\rfloor \text{ or } \left\lceil t \cdot \frac{r}{n} \right\rceil.$$

*Proof.* The proof is again simpler if we assume $k = tr/n$ is an integer. If not, the same argument works controlling the errors. In that case, let $k_1 = tr/n + i$ and $k_2 = tr/n + i + 1$ and as before, let $r = c_r n$ and $t = c_t n$. Note that

$$\frac{\frac{\binom{r}{k_1}\cdot\binom{n-r}{t-k_1}}{\binom{n}{t}}}{\frac{\binom{r}{k_2}\cdot\binom{n-r}{t-k_2}}{\binom{n}{t}}} = \frac{\binom{r}{k_1} \cdot \binom{n-r}{t-k_1}}{\binom{r}{k_2} \cdot \binom{n-r}{t-k_2}} = \frac{\binom{c_r n}{c_t c_r n+i} \cdot \binom{(1-c_r)n}{(1-c_r)c_t n-i}}{\binom{c_r n}{c_t c_r n+i+1} \cdot \binom{(1-c_r)n}{(1-c_r)c_t n-i-1}} = \frac{(c_t c_r n + i + 1) \cdot ((1-c_t)(1-c_r)n + i + 1)}{((1-c_t)c_r n - i) \cdot ((1-c_r)c_t n - i)}$$

If $i \geq 0$, then the last term is at least $\frac{(c_t c_r n)\cdot((1-c_t)(1-c_r)n)}{((1-c_t)c_r n)\cdot((1-c_r)c_t n)}$ which is greater than one. If $i \leq -1$, then the last term is $\frac{(c_t c_r n)\cdot((1-c_t)(1-c_r)n)}{((1-c_t)c_r n)\cdot((1-c_r)c_t n)}$ which is smaller than one.

Thus, $\frac{\binom{r}{k}\cdot\binom{n-r}{t-k}}{\binom{n}{t}}$ is increasing as $k$ increases up to $tr/n$ and then decreases. Thus, the maximum is reached at $tr/n$. $\square$

*Proof of Lemma 4.1.* We first use a simple counting argument to obtain an expression for $\Pr\left(\mathcal{E}_{k,t,r,n}\right)$ as a ratio of binomial coefficients. We note that there are $\binom{r}{k}$ collections of $k$ red balls, $\binom{n-r}{t-k}$ collections of $t-k$ blue balls, and that the total number of collections of $t$ balls is $\binom{n}{t}$. Since the $t$ balls are drawn uniformly at random without replacement, we get

$$\Pr\left(\mathcal{E}_{k,t,r,n}\right) = \frac{\binom{r}{k} \cdot \binom{n-r}{t-k}}{\binom{n}{t}}.$$

636 The $O(1/\sqrt{n})$ bound now follows directly from Lemma E.2 and Lemma E.3. $\qquad\square$

637 Next, we prove Lemma 4.3.

638 **Lemma 4.3.** *Fix any deterministic algorithm $\mathcal{A}$. For any $t = \Theta_1(n)$, $\mathbb{E}\left[\mathsf{cost}_\mathcal{A}(t)\right] = \Omega\left(\sqrt{n}\right)$.*

*Proof.* Let $\sigma$ be a permutation of $[t]$ and $\mathcal{O}_\sigma$ the event that $a_{\sigma(1)} < a_{\sigma(2)} < \ldots < a_{\sigma(t)}$. For any deterministic algorithm $\mathcal{A}$, the choice of the position $\pi(a_t)$ where to place the $t$-th element depends only on $\sigma$. Let $k = \sigma^{-1}(t)$ be the relative rank of the $t$-th element. Since the distribution of $k$ is uniform in $[t]$ (see the proof of Proposition 3.3), then we have that:

$$\Pr\left[\frac{t}{4} \le k \le \frac{3t}{4}\right] = \frac{1}{2}$$

Conditioned on that event $k = \Theta_1(t)$ so we are in the conditions of Lemma 4.2. Therefore, the probability of each rank given the observations is at most $O(1/\sqrt{n})$. Therefore, there is a constant $c$ such that:

$$\Pr\left[\left|\mathsf{rk}(a_{(t)}) - \pi(a_{(t)})\right| < c\sqrt{n} \;\middle|\; \frac{t}{4} \le k \le \frac{3t}{4}\right] \le \frac{1}{2}$$

639 Finally, we observe that:

$$\begin{aligned}
\mathbb{E}\left[\mathsf{cost}_\mathcal{A}(t)\right] &\ge \frac{1}{2} \cdot \mathbb{E}\left[\left|\mathsf{rk}(a_{(t)}) - \pi(a_{(t)})\right| \;\middle|\; \frac{t}{4} \le k \le \frac{3t}{4}\right] \\
&\ge \frac{1}{2} \cdot c\sqrt{n} \cdot \Pr\left[\left|\mathsf{rk}(a_{(t)}) - \pi(a_{(t)})\right| \ge c\sqrt{n} \;\middle|\; \frac{t}{4} \le k \le \frac{3t}{4}\right] \ge \frac{c\sqrt{n}}{4}.
\end{aligned}$$

640 $\qquad\square$

641 We are now ready to prove Theorem 2.

642 *Proof of Theorem 2.* For any deterministic algorithm, sum the bound in Lemma 4.3 for $\Theta(n)$ time
643 steps. For randomized algorithms, the same bound extends via Yao's minimax principle. The reason
644 is that a randomized algorithm can be seen as a distribution on deterministic algorithms parametrized
645 by the random bits it uses. If a randomized algorithm obtains less than $O(n\sqrt{n})$ inversions in
646 expectation, then it should be possible to fix the random bits and obtain a deterministic algorithm
647 with the same performance. $\qquad\square$

# F   Hardness of Online Ranking with Adversarial Ordering

649 **Proposition F.1.** *If the ordering $\sigma$ of the arrival of elements is adversarial, then any algorithm has*
650 *cost $\Omega(n^2)$ in expectation.*

651 *Proof.* At a high level, we construct an ordering such that at each iteration, the arrived element is
652 either the largest or smallest element not yet observed with probability $1/2$ each. Since the algorithm
653 cannot distinguish between the two cases, it suffers a linear cost in expectation at each arrival.

654 Formally, we define $\sigma$ inductively. At round $t$, let $i_{t,-}$ and $i_{t,+}$ be the minimum and maximum
655 indices of the elements arrived previously. We define $\sigma(t)$ such that $\sigma(t) = a_{i_{t,-}+1}$ with probability
656 $1/2$ and $\sigma(t) = a_{i_{t,+}-1}$ with probability $1/2$. Thus, the $t$th element arrived is either the smallest or
657 largest element not yet arrived.

The main observation is that the pairwise comparisons at time $t$ are identical whether $a_{(t)} = a_{i_{t,-}+1}$ or $a_{(t)} = a_{i_{t,+}-1}$. This is since all the elements previously arrived are either maximal or minimal and there is no elements that are between $a_{i_{t,-}+1}$ and $a_{i_{t,+}-1}$ that have previously arrived. Thus the decision of the algorithm is *independent* of the randomization of the adversary for the $t$th element. Thus for any learned rank at time $t$, in expectation over the randomization of the adversary for the element arrived at time $t$, the learned rank is at expected distance of the true rank at least $n/4$ for $t \le n/2$. Thus the total cost is $\Omega(n^2)$ in expectation. $\qquad\square$