[Reviews · NeurIPS 2019]

Reviewer 1



------------------ Post rebuttal comment: The authors have addressed all my comments and, in my view, have also responded satisfactorily to the comments made by the other reviewers. I will keep my score as a 7 as I consider that the paper will be of interest to a small fraction of the NeurIPS community. --------------- a) summary of the content: In this paper the authors consider a generalization of the classical secretary problem where $n$ elements from an ordered set arrive in random order one at a time, and one has to irrevocably assign a rank to each arriving element knowing only the pairwise comparisons with elements which have arrived previously. The objective is to minimize the number of pairs that are inverted with respect to the true order. The authors propose a randomized online algorithm that give $O(n^{3/2})$ inversions in expectation and show that no online algorithms can do better than $\Omega(n^{3/2})$. Upon the arrival of an element, the algorithm works in two steps. The first step estimates the true rank of the arrived element using partial comparisons seen so far. The second step assigned the element to the still available rank closest to this estimated rank. The analysis consists in separately bounding total expected errors due to each of these two steps. It uses a standard Chernoff bound argument for the first step, and a connection to linear probing in hashing for the second step. The lower bound they introduce a new anti-concentration inequality for a generic balls in bins problem with correlated sampling. b) strengths and weaknesses of the submission. * originality: I enjoyed reading this paper. It introduces a new and interesting twist on the secretary problem, thereby providing a stylized theoretical version capturing the main essence of the task of ranking in many online settings. Some part of the analysis also provides some novel techniques that may be independently useful for other purpose (e.g. the new anti-concentration inequality). The proposed randomized algorithm is natural and somewhat unsurprising, but its analysis building upon connections to linear probing is interesting. * quality: By in-large the paper seems to be technically sound. I have gone through most proofs in detail, and although some would benefit with added clarity (see some examples below), I haven't found any main flaws. * clarity: The paper is in general well written, although there is room for some improvement. I list some examples below. * significance: In terms of significance, I believe that this work would be of interest to a small fraction of researchers within NIPS. In fact, in terms of fit, this looks like more a submission to be found within SODA. * other details/comments: - p.2, line 42: the optimal => an optimal \- p.4, line 155 to 159: this is a bit of a repeat, and somewhat ill-placed. Should appear before line 148 \- p.4, measure of sortedness: perhaps indicate whether there are other interesting measures one could consider (besides Kendall' tau and Spearman's footrule) \- p.5, line 192,1: $R$ should be defined (set of available rank); $r_1$ should be defined as 0 \- p.5, line 192,5: how are ties within the argmin dealt with? \- p.5, line 203: this equation is not numbered, but seems to be referred to as Eq. (3.2) later on; so it should be numbered. Same elsewhere where you have an equation which you want to refer to later on ... \- p.5, proof of Proposition 3.3: in the introduction of the event ${\cal O}_\sigma$ 5 lines above, it was for a permutation $\sigma$ on $t$ elements. Now in the proof, $\sigma$ is a permutation over $t-1$ elements. Which conditional event do we have in the conditional probability. By the way that formula indicates that the relative rank $r_t$ is {\em conditionally} uniformly distributed ... \- p.5, line 227: $t$-the should be $t^{th}$. Also what bound do you refer to at the end of the sentence? \- p.7, line 247: shouldn't we say: is popular at time $t$ w.p. at most $e^{-\Omega(\alpha)}$, that would help in the proof of Lemma 3.5, otherwise I don't see how the last equality in 263 can be true as the integral could be infinite? \- p.8, line 277: should $O(n\sqrt{n})$ be $\Omega(n\sqrt{n})$?

Reviewer 2



This paper considers the problem of reconstructing a random permutation online. Similar to the secretary problem, n items arrive online and the algorithm is given each item's rank relative to items that arrived before. The algorithm must then guess the item's absolute rank among all n. The goal is to minimize the distance to the true ranking, as measured by the total number of inversions. The authors consider a method based on guessing each item's rank by assuming the ranks seen thus far are uniformly spread (plus some random noise). They show this method results in O(n^{3/2}) inversions in expectation. They moreover show that no algorithm can do asymptotically better. This is an interesting and non-trivial probabilistic exercise, with interesting connections to the theory of random search trees. The authors do a good job of explaining the technical challenges and how they overcome them. But I found it hard to get excited about the results. Since the problem is so difficult, and the total number of inversions is quite high even if the algorithm does "well," it is hard to understand whether this analysis separates good algorithms from bad algorithms. Indeed, the general approach used here is the first thing one would try. Is there a different natural approach that does not work as well? The one especially interesting aspect of the algorithm is the addition of random noise for the analysis, which is useful in other learning contexts as well. Is this crucial? I.e., does the algorithm perform worse without using noise, or is it just helpful for the analysis? Overall, while I appreciate the technical analysis behind this work, I am not sure what insight to take away from the paper. For that reason, I think the paper is not quite ready for such a competitive venue. Comments post-rebuttal: I take the authors' point that whether the "total number of inversions is high" depends on the normalization. But I don't think renormalizing affects the question of whether the proposed approach is the "right" one for any particular application. I feel as though this could be addressed with some compelling use-cases, to illustrate why "total number of inversions" is the right objective function. Regardless, I found it interesting that bucketing techniques get only part of the way to the improved performance, and this helped clarify for me what is technically challenging about this analysis. Post response and discussion, I feel as though this is a cool and clever mathematical exercise, but without a concrete application or a crisp take-away message, I still worry about fit.

Reviewer 3



EDIT after rebuttal: Overall, I agree that this should be viewed as a theory/algorithm paper; on that note I thought having the same technique apply to more than one related model would definitely help the paper. The authors did say in the rebuttal that their technique extends to two other extensions they consider. I think that definitely makes this paper technically stronger. In terms of fit, my initial impression was that this could be useful in applications of learning to rank when more information is available or can be obtained cheaply; thus becoming an online combinatorial optimization problem (the typical approach seems to assume bandit feedback). Also, since the algorithm is easy to implement, I assumed that practicality of the algorithm is not much of a concern. But given that I have personally not worked with such applications, I would refrain from making stronger claims. I just chose to give the paper the benefit of doubt. Nonetheless, the authors could justify the fit better for the reader by coming up with a concrete application and test their algorithm (some of these prior work that uses bandits seem to have them) in the future version. Post discussion/rebuttal I would still hold my opinion that this is a weak accept; there is something in this paper for a small segment of this community. Just not sure if it will be more broadly relevant. ------ This paper considers a generalization of the secretary problem called the secretary ranking problem. We have n elements coming online in a random permutation order with inherent ordering amongst them. When an element i arrives, we have an oracle that gives results of pairwise comparison of element i with all elements that have arrived so far. Based on this information, the goal is to assign a rank to element i which is irrevocable. After all the n elements have arrived, the penalty paid by the algorithm is the total number of inversions in the assigned ranks. The authors device an algorithm that is tight and obtains a $O(n \sqrt{n})$ number of inversions. The algorithm is simple'' in the sense that it is easy to implement (though highly non-tricial). The analysis proceeds in a fairly natural way connecting well-known concepts. Though the authors claim that they make an interesting connection to linear probing in hashing, it is a fairly straightforward connection in my opinion, given the algorithm which is non-trivial. The authors also prove a matching lower-bound which proceeds by first proving a new anti-concentration theorem for a specific balls and bins scenario via counting (as is the common approach for proving concentration results in balls and bins type problems). Overall, I think the problem is interesting and am convinced that it has many applications. Given this, this paper is a good start and provides a complete understanding for the exact model proposed. However, my main criticisms are the two points below, on the model itself, which would be natural features when applying this to real-world applications. One downside is that the authors assume access to *exact* pairwise comparisons; this may not be true with many real-world applications. I suspect that their algorithm and analysis would potentially extend to noisy pairwise comparison after making sufficiently *nice* assumptions about the nature of the noise. I urge the authors to explore this direction and/or comment if this follows from their algorithm and analysis. Another downside of this model is that the algorithm is assumed to have pairwise comparison with every element that has arrived so far. Thus, overall the comparison oracle is still making $O(n^2)$ comparisons. An extension where the algorithm can choose any number of the previously arrived elements and only compare with those and the goal is to obtain $O(n^{3/2})$ inversions while minimizing the number of comparisons is an interesting direction to consider. Though this is a theory paper, I would have liked to see some experiments on a real-world application. Even if real-world datasets are not easy to come by that matches the problem setting, running some simple simulations should be useful for readability (and also quickly convincing the reader that the proof on the number of inversions is indeed correct). I would urge the authors to consider doing this in the final version of the paper. In summary, here are the pros and cons of this paper. Pros: - Interesting extension of the classical secretary problem - Model has many applications - Optimal algorithm with matching upper and lower-bounds. - Well-written paper and enjoyable to read. Cons: - Model is a good start but does not fully account for issues in real-world applications. Two important additions could be noisy comparison, and limiting the number of comparisons. - Lack of simulations or experiments that would be useful. - The analysis of the upper-bound is not particularly challenging.

Reviewer 4



The problem seems closely related to online algorithms for weighted bipartite matching, such as Kesselheim, Thomas, Klaus Radke, Andreas Tönnis, and Berthold Vöcking. "An optimal online algorithm for weighted bipartite matching and extensions to combinatorial auctions." In European Symposium on Algorithms, pp. 589-600. Springer, Berlin, Heidelberg, 2013. This seems to be more general than the problem considered in the submission, but potentially encompasses the secretary ranking problem. Thus, the novelty of the paper may be somewhat marginal. Overall, the paper is clear and well-written. I didn’t read the proofs carefully, but the proof sketches were helpful and convincing.

[Author Response · NeurIPS 2019]

Thank you to all three reviewers for their thoughtful reviews. Please find below responses to specific comments.

**Reviewer 1.**  Thank you for the overall positive review.

- *lines 155 to 159: this is a bit of a repeat, and somewhat ill-placed:* We will modify as suggested.

- *indicate whether there are other interesting measures one could consider:* Another measure is the distance to monotonicity, which corresponds to the difference between $n$ and the longest increasing sequence. Considering other measures of sortedness is an interesting direction for future work.

- *R should be defined; $r_1$ should be defined as $0$:* We will add a line in the algorithm for initializing $R$. By definition of $r_t, r_1 = 0$ since there is no $t' < 1$.

- *line 192 how are ties within the argmin dealt with?* Ties for argmin can be broken arbitrarily.

- *In the introduction of the event $\mathcal{O}_\sigma$, ...:* The conditional event in the probability is for a fixed permutation over $t - 1$ elements. We will fix the text to "conditionally uniformly distributed" as suggested.

- *line 227: what bound do you refer to at the end of the sentence?* The bound refers to the expression for the conditional probability between lines 223 and 224, we will make this clearer.

- *line 247: shouldn't we say: is popular at time $t$ w.p. at most $e^{-\Omega(\alpha)}$?* Yes, this should be $\Omega$ instead of $O$.

- *line 277: should $O(n\sqrt{n})$ be $\Omega(n\sqrt{n})$?* Yes, we will fix this, thank you for catching these typos.

**Reviewer 4.**  It seems that a main concern is that "the total number of inversions is quite high" which makes it "hard to get excited about the results." A better way to look at the result is to see what fraction of the secretaries are incorrectly ranked with respect to a randomly picked secretary. By allocating at random, each secretary is incorrectly ranked with respect to $1/2$ fraction of remaining secretaries. With our algorithm, each secretary is incorrectly ranked with respect to only a $1/\sqrt{n}$ fraction of remaining secretaries.

- *Is there a different natural approach that does not work as well?*
  One of our first attempts to this problem was an inductive bucketing algorithm where we use the first $s$ elements to partition the positions within buckets in which we add future elements. The non-inductive and inductive bucketing algorithms obtain $O(n^{9/5})$ and $O(n^{5/3})$ inversions respectively.

- *Does the algorithm perform worse without using noise, or is it just helpful for the analysis?*
  This is a good question. We do not know what is the number of inversions obtained by the algorithm without noise, analyzing this algorithm would require new non-trivial techniques. Adding noise is a central idea needed for our analysis.

**Reviewer 5.**  The two additions to the model that are mentioned in the review are very interesting questions that indeed provide a closer account for issues in real-world applications. We can extend our results to account for both of these additions. In the final version of the paper, we will add these additional results. We do also note that this is the first paper studying the secretary ranking problem and that we believe there is value in providing a simple model to start with and to which further additions can then be added.

- Improving the number of comparisons: It is possible to obtain $O(n^{3/2})$ inversions with $O(n \log n)$ comparisons, which is optimal up to lower order terms. This is because the algorithm maintains an ordering of the elements seen so far. Thus, when a new element arrives, the algorithm can find the position of this new element in the current ordering by binary search.

- Noisy pairwise comparisons: we can extend our results to a noisy model where each comparison is, independently, correct w.p. $1 - \epsilon$ and wrong w.p. $\epsilon$. The impact of the noise in the analysis is an additional $\epsilon n$ additive term per element in the estimation cost and no additional term for the assignment cost. The total cost is then $O(n^{3/2} + \epsilon n^2)$. The reason this can be done is that the analysis is modular and decomposes the error in an estimation and assignment terms. If we estimate the new rank using line 3 of Algorithm 1 but with noisy comparisons, we still have an unbiased estimator of the rank (but with larger variance) and Proposition 3.3 still holds. Those are the only requirements for the analysis of the assignment cost, so it remains unchanged.

- If we have noisy comparisons and also want to improve the number of comparisons, the technique in (a) for reducing the number of comparisons no longer holds, but we can still reduce the number of comparisons by estimating $\tilde{r}$ in line 3 of Algorithm 1 by sampling only a few of the previous elements and paying the corresponding cost in increased variance. The same analysis as in (b) still holds.

[Meta-Review · NeurIPS 2019]

Overall, the reviewers liked the setting introduced by this paper, and in particular the associated analysis. The main concern was the fit for NeurIPS. After several discussions, the program committee found that this paper is indeed interesting to a part of the NeurIPS community. The reviewers also discussed the concern about relations to online matching, but found that the paper is sufficiently different. I urge the authors to add to the paper to address the reviewer comments, and possible confusions. Consider adding discussions, examples and conclusions. Another point that came up were algorithmic implications, please add to this if possible.